# Genome-Wide DNA Methylation Profiles in Whole-Blood and Buccal Samples—Cross-Sectional, Longitudinal, and across Platforms

**DOI:** 10.3390/ijms241914640

**Published:** 2023-09-27

**Authors:** Austin J. Van Asselt, Jeffrey J. Beck, Casey T. Finnicum, Brandon N. Johnson, Noah Kallsen, Jouke Jan Hottenga, Eco J. C. de Geus, Dorret I. Boomsma, Erik A. Ehli, Jenny van Dongen

**Affiliations:** 1Avera McKennan Hospital, University Health Center, Sioux Falls, SD 57105, USA; austin.vanasselt@avera.org (A.J.V.A.);; 2Department of Biological Psychology, Vrije Universiteit, 1081 HV Amsterdam, The Netherlands

**Keywords:** DNA methylation, epigenetics, Illumina microarrays, tissue comparison, platform comparison, longitudinal design

## Abstract

The field of DNA methylation research is rapidly evolving, focusing on disease and phenotype changes over time using methylation measurements from diverse tissue sources and multiple array platforms. Consequently, identifying the extent of longitudinal, inter-tissue, and inter-platform variation in DNA methylation is crucial for future advancement. DNA methylation was measured in 375 individuals, with 197 of those having 2 blood sample measurements ~10 years apart. Whole-blood samples were measured on Illumina Infinium 450K and EPIC methylation arrays, and buccal samples from a subset of 58 participants were measured on EPIC array. The data were analyzed with the aims to examine the correlation between methylation levels in longitudinal blood samples in 197 individuals, examine the correlation between methylation levels in the blood and buccal samples in 58 individuals, and examine the correlation between blood methylation profiles assessed on the EPIC and 450K arrays in 83 individuals. We identified 136,833, 7674, and 96,891 CpGs significantly and strongly correlated (>0.50) longitudinally, across blood and buccal samples as well as array platforms, respectively. A total of 3674 of these CpGs were shared across all three sets. Analysis of these shared CpGs identified previously found associations with aging, ancestry, and 7016 mQTLs as well.

## 1. Introduction

Epigenetic mechanisms provide a molecular link from an individual’s environmental exposures to their expressed phenotype through the modification of gene expression. Epigenetic marks are mitotically, and sometimes meiotically, heritable elements that regulate gene expression without modifying the underlying DNA sequence [1]. DNA methylation, one of the most studied epigenetic mechanisms, is the addition of a methyl group onto cytosine nucleotides at the C5 position forming 5-methylcytosine, typically next to a guanine (forming a CpG site), and this modification can regulate gene expression and mediate downstream effects of genetic variants [2]. DNA methylation has become of particular interest in epigenetic research as it is the easiest to assess from a genome-wide perspective in large numbers of individuals due to the scalability of the laboratory practices required to investigate it, typically by array-based assays. DNA methylation can be utilized to quantitate cellular differences, environmental influences on a particular trait, and changes in cell senescence that occur during aging [2,3]. The availability of easily accessible tissue samples (whole blood, buccal, saliva, etc.) with these scalable technologies allows for the implementation of large-scale epigenome-wide association studies (EWASs), which may lead to the development of biomarker assays that can serve as risk indicators for disorders and disease, monitor treatment effects, and predict treatment outcomes. Though the increase in DNA methylation research has been substantial, there are many questions that remain regarding the reliability, application, and extrapolation of this information across tissue, over time, and across measurement platforms.

Studies investigating DNA methylation in relation to individual age and mortality have led to some of the biggest breakthroughs in the field over the past decade. Previous research investigating DNA methylation in relation to chronological age identified one of the strongest, measurable associations that can accurately predict an individual’s chronological age [4]. Since an initial study by Horvath et al. in 2013, additional studies investigating DNA methylation in groups of individuals of differing ages have identified even more associations with DNA methylation [5,6,7]. There is a multitude of studies that have investigated the association of age and DNA methylation, but there are remarkably few longitudinal projects, i.e., with the inclusion of the same individuals over time for the investigation of CpGs. Likely, this is partly due to the time consumption inherent to such study designs [4]. Understanding the stability (or lack thereof) of the entire DNA methylation profile via individual CpG measurements over time will quantify the susceptibility to a change in the methylome and offer a starting point to further investigate these CpGs longitudinally.

The epigenetic mechanisms that drive cellular differentiation have been studied extensively [8]. Large efforts have been put forth to quantitate and identify the similarities and differences in DNA methylation across tissues [9]. For example, Slieker et al., using an Illumina 450K array, reported CpGs with correlated methylation levels between blood and 16 different tissues, including the brain, buccal, liver, kidney, subcutaneous fat, monocytes, and T-helper cells [10]. In this comparison, they included 96 blood/buccal samples and were able to find 4857 differentially methylated positions (DMPs), of which 126 were also identified in the 426 DMPs found in whole blood, showing modest overlap between the two sample types [10]. Hannon et al. compared DMPs in DNA derived from whole blood to DMPs in DNA isolated from targeted regions in the brain, which has resulted in the creation of a number of online resources that allow researchers to compare methylation levels on a per-CpG basis between two tissue sources [11]. A previous study conducted by Braun et al. compared the methylation levels of samples from multiple tissue sources (which included whole blood and buccal) in 27 individuals and calculated an overall correlation of DNA methylation across subjects of 0.81 comparing the methylation levels of whole-blood and buccal samples [12]. Adding to the previous findings in this field can improve the understanding of extrapolating findings across different tissues.

The Illumina Infinium 450K methylation array was introduced just over a decade ago in 2012 and has been one of the most widely utilized tools to investigate DNA methylation [13,14]. Subsequently, the Illumina Infinium EPIC methylation array was released as the successor to the 450K array in 2016, covering over 90% of the CpGs on the 450K array while also interrogating another ~400,000 CpGs [15]. Additionally, further iterations of the EPIC array (such as version 2) have been released recently, which aim to improve its coverage and remove under-performing probes [16]. Due to the continual rollout of these platforms, the EPIC array was created to be compatible with data collected on the 450K array to allow for large, retrospective studies and meta-analyses combining data sets from both. Previous work comparing data from samples measured on 450K and EPIC showed strong concordance for genome-wide methylation status (r = 0.99) [17]. However, recent publications by Sugden et al. and Olstad et al. demonstrated that correlations of methylation levels on a CpG-specific basis across these two platforms varied widely, suggesting that some CpGs are more reliably measured than others [18,19].

Here, we examined CpG-specific longitudinal (~10 years) methylation correlations, cross-tissue methylation correlations from whole-blood and buccal samples, and methylation correlations measured across the two major Illumina platforms (Illumina 450K–Illumina EPIC v1, San Diego, CA, USA) in samples taken from 375 participants in two age groups of roughly equal size (and sex): the first group of ~19 years of age, on average, at the first assessment consisted of twins and siblings, and the second group of ~45 years of age, on average, consisted of their parents. All blood and buccal samples were collected by the Netherlands Twin Register in the 1990s and 2000s.

## 2. Results

For each of the three comparisons (Table 1), we obtained the correlation between DNA methylation levels at individual CpGs. Correlations greater than 0.5 were considered strong. The total number of CpGs investigated in each set of data (Table 1) was utilized for the multiple testing correcting via a Bonferroni correction.

### 2.1. Longitudinal Comparison

Our gee analysis over approximately 10 years (mean, 9.98 years; median, 9.83 years; range, 2.8–15 years) across all CpGs yielded a mean correlation of 0.250 and median of 0.204. At 136,833 CpGs, DNA methylation level was strongly and significantly correlated over time (18%, Table 2). When restricting to the 10% most variable CpGs in this analysis (75,926), 28,919 (38.1%) were identified to have significant, strong correlations over time. The mean correlation of the 10% most variable was r = 0.50, with the median being r = 0.53. Appendix A contains a list of the 136,833 CpGs and additional information regarding them.

### 2.2. Cross-Tissue Comparison

Our gee analysis across all 759,263 CpGs on the EPIC array yielded a mean correlation of 0.071 and a median of 0.067. We identified 7674 CpGs with strong (r > 0.50) and significant correlations (1%, Table 2). Next, we investigated the overlap of the 10% most variable CpGs in whole-blood and buccal samples. The overlap of variable CpGs in blood and buccal samples was modest: 22,459 (out of 75,926 possible, 29.6%). The most variable CpGs in blood were also among the top 10% most variable CpGs in buccal samples. When restricting to these shared 10% most variable CpGs that overlap in this analysis (22,459), 1691 (7.5%) were identified to have a significant, moderate correlation across blood and buccal samples. The mean correlation of the shared 10% most variable was r = 0.27, with the median being r = 0.14. A list of the 7674 CpGs and Appendix A regarding them can be found in Appendix A.

### 2.3. Cross-Platform Comparison

Our gee analysis across all 386,805 CpGs that overlap across the EPIC and 450K platforms yielded a mean correlation of 0.287 and a median of 0.197. We identified 96,891 CpGs with strongly and significantly correlated methylation levels across platforms (25%, Table 2). Next, we investigated the overlap of the 10% most variable CpGs between the two array platforms and identified that 31,546 of the potential 38,681 CpGs (81.6%) were of the 10% most variable for both arrays. When restricting to the 10% most variable CpGs shared by both the EPIC and 450K array in this analysis (31,546), 24,518 (77.7%) were identified to have a significant, strong correlation across array platforms. The mean correlation of the 10% most variable was r = 0.83, with the median being r = 0.86. Appendix A contains a list of the 96,891 CpGs and their Appendix A.

Next, we compared our cross-platform results with results from a previously published study that investigated the correlation between repeated measures of 350 blood samples on the 450K and EPIC v1 array (Sugden et al.) [18]. Utilizing a similar threshold of 0.5 to maintain consistency, we found that Sugden et al. identified 79,007 CPGs with an ICC (intra-class correlation) value above this threshold [18]. Of these, 56,930 (72.1%) were also identified in our data to have a significant, strong correlation across platforms [18]. When comparing in the group of shared highly correlated/reliable CpGs, our gee estimates and the ICCs generated by Sugden et al. closely resemble one another (r = 0.544, correlation comparing the gee estimates and ICCs) [18]. This similarity is even greater when comparing across all 386,805 CpGs (r = 0.810) [18]. Appendix A shows the relationship of these two measurements for the 386,805 CpGs that overlap.

### 2.4. Overlap of Reliable CpGs Significantly and Strongly Correlated across Time and Tissue

Investigating our three sets of results from the longitudinal, across-tissue, and across-platform analyses, we identified 3674 CpGs with correlations >0.50 that also reached significance. These CpGs yielded a mean correlation of 0.87 and a median of 0.89. A list of the 3674 CpGs and additional information regarding them can be found in Appendix A. Figure 1 visualizes the overlap of the CpGs identified in each analysis. Appendix A provide additional information on the overlap of significant and strongly correlated CpGs for each of the three analyses.

### 2.5. EWAS Atlas and BBMRI mQTL Database Query

A query of the EWAS Atlas database providing the 3674 CpGs that were identified to overlap between all three experiments highlighted several phenotypes that are associated with hundreds of these CpGs (Table 3) [20]. The EWAS Atlas database can be accessed via https://ngdc.cncb.ac.cn/ewas/atlas/index (accessed on 5 April 2023). Most notably, ancestry, gender, and aging were previously associated with 328, 195, and 191 of these CpGs, respectively [20]. Figure 2 highlights the genomic locations with the greatest associations with these 3674 CpGs.

A query of the BBMRI mQTL database with the CpGs with a significant, strong estimate identified in each of the three analyses, along with the 3674 overlapping CpGs, was performed [21]. The BBMRI database can be accessed via https://bbmri.researchlumc.nl/atlas (accessed on 16 February 2023). In all four comparisons, we identified an increased prevalence of CpGs associated with a known mQTL compared to the standard genome-wide prevalence rate put forth by Bonder et al. [21]. The results are summarized in Table 4. The group of 3674 overlapping CpGs used to query BBMRI yielded 1989 CpGs associated with 7016 mQTLs (mean = 3.5 mQTLs per associated CpG, median = 3) [21]. The longitudinal CpGs used to query BBMRI totaled 69,570, with 42,881 of those CpGs associated with 121,253 mQTLs (mean = 2.8 mQTLs per associated CpG, median = 2) [21]. The cross-tissue CpGs used to query BBMRI totaled 4408, with 2181 CpGs associated with 7402 mQTLs (mean = 3.4 mQTLs per associated CpG, median = 3) [21]. The platform CpGs used to query BBMRI totaled 96,891, with 61,503 CpGs associated with 165,484 mQTLs (mean = 2.7 mQTLs per associated CpG, median = 2) [21].

### 2.6. Investigation of Individual CpGs of Interest

To make a more nuanced investigation of these data, we investigated the correlations of standard beta values for two specific CpGs: cg05575921 (gene *AHRR*) and cg16867657 (gene *ELOVL2*), which are known to be associated with smoking status and chronological age, respectively. Cg05575921 showed correlations of 0.82, 0.52, and 0.96 when assessed longitudinally, between blood and buccal samples as well as across platforms, respectively. Cg16867657 showed correlations of 0.92, 0.51, and 0.91 when assessed longitudinally, between blood and buccal samples as well as across platforms, respectively. These comparisons are highlighted in Figure 3 and Figure 4.

## 3. Discussion

The collective knowledge of epigenetics has expanded rapidly over the past two decades. With this rapid expansion, the need has risen dramatically to assess the validity of expanding, combining, and extrapolating results across tissues, biological age, and measurement platforms. Here, we validate previous findings, expand on the current understanding of the biological influences of DNA methylation data captured via Illumina microarrays, and offer explanations of these influences and accompanying recommendations.

### 3.1. Longitudinal Comparison

In total, we identified 136,833 CpGs that showed a strong, significant correlation in longitudinal samples collected, on average, 10 years apart. Changes in DNA methylation as one ages have been proven to have moderate to strong correlations with chronological age [22]. Further, combining this association with the age of several hundred of these CpGs has yielded some of the strongest measures of biological function by DNA methylation [23]. Many of the previously generated epigenetic clocks are calculated by only a few hundred CpG sites [22,23]. Though this involvement with time is present, a large portion of measurable CpG sites appear quite stable over time, as shown here (at least relative from individual to individual). Previous findings have shown that, while DNA methylation is dynamic and is influenced by aging, methylation profiles typically remain fairly stable (when compared to other individuals) over short periods of time, and it appears difficult to modify the relationship between DNA methylation and aging, though not impossible, as recent findings have suggested [7,24,25]. This stability is also captured here, showing that a large portion of CpGs correlate extremely well over time. Of those CpGs that did not reach the threshold of 0.50 utilized here, many still show modest positive correlations. Further, we identified that the CpGs showing the largest variance in beta value also generally showed a much greater correlation compared to the genome-wide average. This indicates that the CpGs that correlate well over time, while relatively stable, show significant inter-individual variability, and this variability appears to be driven by influences outside of the aging process. We should note that CpGs with a strong correlation across time points can still show changes with age, though these changes are consistent when compared to other individuals. For example, an individual with a higher level at time point 1 (compared to other individuals) will also tend to have a high level at time point 2 (compared to other individuals).

### 3.2. Cross-Tissue Comparison

We identified 7674 CpGs with a strong and significant correlation across blood and buccal samples (r > 0.50). Tissue-specific differences in DNA methylation are some of the most known and well characterized [10,11,26,27]. Even though epigenetic processes are a large driver of cellular differentiation, we still identified thousands of CpGs across the genome that proved to be consistently methylated in the same individual (relative to other individuals), regardless of the sample source (whole blood vs. buccal). That being said, we found, comparatively, a significant reduction in the amount of correlated CpGs in this assessment versus our assessments longitudinally and across platforms, which was to be expected. Further, a significant majority of the CpGs identified across sample sources largely overlapped with CpGs that correlated strongly over time (Figure 1). When assessing the variability of each CpG in both whole-blood and buccal samples independently, we found that they shared 29% of these CpGs, indicating that while some sample variation appears consistent across specimen sources, a large majority of it is unique to the source. The 7674 CpGs found to be related between these two sample types could provide significant value to future studies looking to investigate the utilization and comparison of whole-blood and buccal samples as surrogate tissues for investigating various diseases and phenotypes. Further, the identification of these 7674 CpGs could allow for studies aimed at combining DNA methylation data from whole-blood and buccal samples utilizing strictly these CpGs. Both cg05575921 and cg16867657, two CpGs previously identified to be associated with smoking and aging, for example, fall into this category of CpGs with strong correlations across whole-blood and buccal samples.

### 3.3. Cross-Platform Comparison

Here, we identified 96,981 CpGs (of the 386,805 CpGs included) to have a significant and strong correlation value when compared across the EPIC and 450K arrays (25%). Similar to previous findings, we found that a large majority of CpGs do not correlate well when comparing arrays. Interestingly, however, a similar phenomenon has been shown in replicate samples assessed twice on the same platform (for example, the same sample assessed twice on the EPIC array), demonstrating that the results are not specific to comparing the two different platforms [18]. Though a majority of CpGs did not achieve a strong estimate, a large portion resulted in low to moderate positive estimates.

Importantly, similar to previous findings, we identified that the per-CpG correlation level is heavily influenced by individual CpG variability [28,29]. Our data showed that the shared 10% most variable CpGs had an average estimate of r = 0.83, and the 96,981 CpGs identified to have a strong, significant estimate had a greater average standard deviation in beta value than that of the genome-wide average (0.05 vs. 0.03, respectively). This indicates to us that the array, while highly accurate, cannot potentially provide the level of nuance and sensitivity at these loci necessary to discern the minute differences in methylation level that may or may not be present. Further, in EWAS analyses where statistical power is often an issue, researchers may consider restricting their analysis to the set of CpGs with the largest variability and reliability. Restricting to this set of CpGs may also benefit the construction of more accurate epigenetic scores.

To further compare our findings to previous results, we performed a comparison of the strong, significant CpGs we identified to the CpGs identified by Sugden et al. to have an intra-class correlation coefficient (ICC) > 0.50 [18]. Generally, overall, with 386,805 overlapping CpGs, our estimate values coincide nicely with the ICCs calculated by Sugden et al. (r = 0.810) [18]. We identified a comparable number of strongly correlated (r > 0.5) CpGs (96,981 in our study; 79,007 by Sugden et al.) calculated to meet the threshold of 0.50 [18]. Lastly, the average correlation across all genome-wide CpGs was similar (r = 0.287 in our study; mean ICC = 0.212 by Sugden et al.) [18].

### 3.4. Interrogation of the Overlapping CpGs

When cross-comparing our findings of the previously described experiments above, we identified 3674 CpGs that show significant, strongly correlated methylation levels. These overlapping CpGs are particularly interesting because they show some form of stability over time and between cells of different types even though these are two of the largest known influencers of DNA. The fact that the CpGs identified here demonstrate dynamic differences in methylation status between individuals but also relative stability in relation to cell type and age, their presence may hint at another physiological mechanism driving their methylation status (such as genetic variation).

### 3.5. EWAS Atlas and BBMRI

As previously stated, two of the largest known contributors to differences in DNA methylation in humans are the cell type and age of the individual [30,31]. Here, we have identified 3674 CpGs that are consistent across array platforms, highly dynamic in their methylation status between individuals, and consistent in individuals as they age and in multiple cell types. This begs the question of what is driving the variable methylation status of these loci. We first queried the EWAS Atlas database with these overlapping CpGs and identified several phenotypes and measurable traits that have been previously associated with hundreds of CpGs [20]. Most notably, the trait that associated with the largest amount of CpGs was ancestry (328 CpGs) [20]. Ancestry associating most abundantly with these CpGs indicates genetics as a potential driver of these dynamically methylated CpGs. The previous literature has identified a multitude of CpGs driven by mQTLs, indicating that a large portion of measurable CpGs is influenced by genetic variants (~35% of measurable genome-wide CpGs) [21]. After comparing our list of CpGs to a known list of identified mQTLs via the BBMRI mQTL database built off of the results of Bonder et al., we saw that 1989 of 3674 had previously been identified to be influenced by at least one mQTL [21]. Compared to the level of general genome-wide association to mQTLs, this group of overlapping CpGs shows a greater prevalence of mQTLs (54.1% vs. 35.9%), as well as a greater amount of mQTL associations per CpG [21]. This increased prevalence of mQTL association indicates that genetic variants contribute greatly to these CpGs that show stability over time and across whole-blood and buccal samples.

Additionally, we queried the BBMRI mQTL database with each of the three CpG groups identified to strongly, significantly correlate longitudinally, across whole-blood and buccal samples, and across platforms [21]. The group of CpGs that correlated across platforms showed the greatest prevalence of mQTL association of all groups tested with 63.5% of the CpGs being previously identified to associate with at least one mQTL, indicating that genetic variants play a major role in the dynamic methylation of these CpGs that replicate well across the arrays [21]. This also likely reflects that mQTL meta-analyses are better powered to detect mQTL associations for CpGs that are reliably measured.

### 3.6. Individual CpG Assessment

There are certain specific CpGs that have been previously identified to be strongly linked to certain traits and molecular processes. To highlight some examples, we investigated two of these CpGs—cg05575921 and cg16867657—which have been shown to be significantly associated with phenotypes of smoking, biological age, and others [32,33]. When investigating these two CpGs across our three assessments, both appeared to replicate very well across platforms (r = 0.96 and r = 0.91, respectively). Further, both CpGs showed strong correlations across whole-blood and buccal samples and longitudinally over a period of 10 years (r = 0.82 and r = 0.92, respectively). For cg05575921, it has been previously documented that smoking status contributes to the hypomethylation of these loci, which are also shown here (Figure 3). Variability in time spent smoking and changes in smoking behavior likely contribute to some of the increased variability (and reduced correlation) seen in individuals over time as well. This reliability in measurement, combined with the stability of the CpGs between tissues and over time, likely contributes to the clarity of their previously discovered associations. The consistency of their measurement provides added clarity that allows their relationships with a phenotype to be more easily distinguishable.

### 3.7. Limitations

There are limitations to address in this study. First, the individuals included in this study are all from a European population based in the Netherlands. Additional studies utilizing additional cohorts should be completed to investigate these findings in individuals of different backgrounds. Additionally, though we identified significant findings, it would be ideal to perform these same analyses on a larger scale with more individuals. Typically, studies of DNA methylation increase in reliability and reproducibility with large sample populations.

## 4. Materials and Methods

### 4.1. Overview

Participants were from twin families of European descent from the Netherlands. Participants ranged in age from 19 to 57 years old. First, a whole-blood sample was collected from 425 individuals from twin families from the Netherlands in the mid-1990s [34]. During enrollment into a second study, whole-blood and buccal samples were then collected simultaneously in the late 2000s in 233 of these same individuals through the Netherlands Twin Register [35].

DNA was extracted at the time of sample collection or immediately prior to the assessment of methylation. A pilot study was conducted in 2018, as part of which EPIC array data were generated for 80 samples that had been previously measured on a 450K array. The remaining samples were then simultaneously measured on an Illumina Infinium EPIC Methylation array. A total of 83 of the whole-blood samples that were collected at the second timepoint were also assessed on an Illumina Infinium 450K methylation array [36]. In total, 657 samples from 375 participants were assessed for genome-wide methylation. A total of 643 samples from 373 individuals passed all quality control metrics. Correlation analyses were performed on 394 samples (197 pairs) for the longitudinal assessment, 116 samples (58 pairs) for the cross-tissue assessment, and 166 samples (83 pairs) for the cross-platform comparison. All individuals were also genotyped.

The study was approved by the Central Ethics Committee on Research Involving Human Subjects of the VU University Medical Centre, Amsterdam, an Institutional Review Board certified by the U.S. Office of Human Research Protections (IRB number: IRB00002991; Federal-wide Assurance: FWA00017598; IRB/institute codes: NTR 03-180).

### 4.2. Sample Collection

The procedures of whole-blood and buccal sample collection have been previously described [34,36]. Briefly, for buccal sample collection, 16 cotton mouth swabs were individually rubbed against the inside of the individual’s cheek and placed in 15 mL conical tubes. The tubes contained 0.5 mL STE buffer (100 mM sodium chloride, 10 mM Tris hydrochloride (pH 8.0), and 10 mM ethylenediaminetetraacetic acid) with proteinase K (0.1 mg/mL) and sodium dodecyl sulfate (SDS) (0.5%) per swab. Individuals were requested to not eat or drink 1 h prior to sample collection.

### 4.3. DNA Extraction

The genomic material analyzed in this study was extracted and assessed for DNA quantity, quality, purity, and individual identity. The DNA analyzed in this study was extracted from whole-blood and buccal samples using a Zymo Quick DNA mini-prep Kit (Zymo Research, Irvine, CA, USA). Genomic material was quantified via an Invitrogen Qubit Broad-Range Fluorescent Assay (Carlsbad, CA, USA), and sample purity was assessed using standard absorbance metrics via a SpectraMax microplate reader (Molecular Devices, San Jose, CA, USA).

### 4.4. Genotyping

Genotype data were generated using an Illumina Infinium Global Screening Array (GSA) on blood samples for all individuals [37]. Genotype data quality assessment and sample call-rate generation were performed utilizing Illumina GenomeStudio 2.0 software [38]. Sample sex-check confirmation and a test of sample heterogeneity were performed using PLINK v1.90. Identity-by-descent (IBD) sharing and familial relationship assessment was performed utilizing bioinformatic software PLINK v1.90. Briefly, IBD sharing functions by comparing the amount (or proportion) of shared alleles between two individuals across all SNPs being measured [39]. The assumed relationships (generated via genotype data) were then compared to the documented familial relationships in our sample manifest to confirm sample identities. In total, 20 samples were omitted for analysis via the EPIC methylation array because of identity mismatch based on genotype data.

### 4.5. DNA Methylation Assessment

DNA bisulfite conversion was performed utilizing a Zymo EZ-96 DNA Methylation Kit [40]. DNA methylation was assessed using an Illumina Infinium EPIC DNA Methylation Array on all 657 samples at the Avera Institute for Human Genetics [40]. The samples were fully randomized across arrays and run on an Illumina Infinium EPIC methylation array. Additionally, DNA methylation was assessed earlier on 83 of the same samples using an Illumina Infinium HumanMethylation450 array at the human genomics facility (Huge-F), Rotterdam, the Netherlands. Details including quality control of the HumanMethylation450 array data have been described previously [36].

### 4.6. DNA Methylation Data Quality Control

Assessment of DNA methylation data quality of the EPIC array data was performed using two bioinformatic tools. First, DNA methylation data were preliminarily assessed using Illumina GenomeStudio 1.0 software. Here, a relative assessment of independent and dependent sample controls was performed. Additionally, CpG detection percentage was also quantified for each sample.

Second, methylation data quality control and normalization were performed using a quality control pipeline developed and outlined by the Biobank-based Integrative Omics Study (BIOS) Consortium. Sample quality was first assessed using the R package MethylAid (v1.34.0), with default thresholds [41]. Array probe filtering and sample normalization were performed using the R package DNAmArray (v2.0.0). Sample normalization was performed with functional normalization (Fortin et al., 2014) as implemented in DNAmArray [42]. The R package omicsPrint (v1.20.0) was utilized to identify genotypes based on methylation probes to verify sample relationships [43]. Both the function getSex from DNAmArray and the R package meffil (v 1.3.4) were used to identify and confirm sample sex based on X chromosome methylation pattern (Appendix A) [44].

Samples were only kept for further analysis if they passed all five quality criteria defined by the MethylAid thresholds. MethylAid quality control plots are provided in Appendix A. In total, 10 low-performing samples were identified (1.5%) and omitted for not reaching at least one of the thresholds. Of the remaining 647 samples, all samples had matching predicted sexes compared to the assumed sex of the sample. Finally, 4 samples were omitted due to incorrect sample relationships identified via omicsPrint, leaving the total number of included samples at 643 (Appendix A).

The first four control probe principal components were included in the functional normalization process. A scree plot of the PCs generated from the control probe data is shown in Appendix A. Next, the following probe filters were applied: Probes were set to missing (NA) in a sample if they had an intensity value of exactly zero, detection *p*-value > 0.01, or bead count < 3. DNAmArray will also remove any probes that show a success rate below 0.95 across all samples.

Finally, previous studies performed by Zhou et al., in 2017 identified polymorphic and cross-reactive probes that are included on Illumina methylation array platforms. Generally, they recommend an omission of approximately 100,000 probes on the EPIC methylation array. This removal process was carried out using the probemasking() function via DNAmArray [45].

Following these steps, a total of 759,263 methylation sites were included in this study out of the total of 865,859 possible CpGs. Only autosomal methylation sites were considered for downstream analyses, which left a total of 742,442 CpGs included in the final analyses. Appendix A shows the distribution of these CpGs based on their beta value in the form of a density plot.

### 4.7. Statistical Analyses

PCA was performed with DNAmArray prior to and after normalization, and the correlation of the first ten PCs with technical and biological variables (e.g., age, sex, epithelial cell proportion) was computed to check for batch effects and biological correlates of variation in genome-wide methylation patterns. These analyses indicated that normalization successfully reduced variation related to technical factors such as 96-well plate position and the location of the sample on the EPIC array, and that biological factors (cellular composition of samples and sex) are the most important drivers of variation in genome-wide methylation levels (as illustrated by their strong correlation with PC1 and PC2; Appendix A). We did not adjust for cell composition estimates as the main objective was to assess how similar DNA methylation profiles are across measurements by different platforms (EPIC vs. 450K), across two distinct peripheral tissues (blood vs. buccal), and across two time points. The platform comparison (450K versus EPIC) was not affected by cellular composition because the 2 arrays were run on the exact same DNA sample. Several factors can contribute to the resemblance of DNA methylation profiles across the other two comparisons. We only adjusted for technical covariates (sample plate and array row) to rule out technical batch effects as a contributor, and regard all biological sources of variation including cellular composition as factors that may affect the similarity of DNA methylation profiles across tissues and over time. Our approach (not adjusting for any factors other than technical variation) is in line with what previous similar studies have performed [10,11].

### 4.8. Methylation Data Annotation

Genomic annotations were gathered from the EPIC manifest file that is provided by Illumina (MethylationEPIC_v-1-0_B5.csv): locations of CpG islands, ENCODE DNase I Hypersensitive sites (DHSs), ENCODE transcription factor binding sites (TFBSs), open chromatin, FANTOM4 and FANTOM5 enhancers, etc.

### 4.9. DNA Methylation Data Analysis

For each of the three major experiments performed here (longitudinal, cross-tissue, cross-platform), we computed the correlation between methylation levels at individual genome-wide CpGs. First, for each comparison, methylation beta values were corrected for sample plate and array row, and residuals were used as input for the correlation analyses (z-score). Due to the related nature of the samples utilized in this study (twin families), correlation analyses were computed utilizing a generalized estimating equations (GEE) model (via the R package “gee” (v4.13-25)) that corrected for the correlation structure in families. The following settings were utilized: Gaussian link function, 100 iterations, and the “exchangeable” option to correct for the familial structure. Residualized methylation levels were standardized (z-score), which makes the regression coefficient from gee equivalent to a correlation coefficient. Bonferroni correction was applied to correct for multiple testing and identify significant CpGs (0.05/total number of CpGs tested). A threshold of 0.50 was utilized to identify CpGs with a strong positive correlation.

As an additional measure to ensure the validity of the gee results, we also generated Pearson’s correlations for each CpG, which were strongly correlated with the correlations obtained in gee (longitudinal r = 0.998, cross-tissue r = 0.914, cross-platform r = 0.996; Appendix A).

For the cross-platform experiment, we wanted to investigate the similarity of our findings to those previously reported by Sugden et al. [18]. We applied an identical threshold of 0.50 to their reported intra-class correlation coefficients (ICCs) to identify CpGs to compare against our results [18].

We reported results for genome-wide CpGs and for the 10% most variable CpGs in blood samples and buccal samples. Finally, we identified CpGs that showed strong, significant (after Bonferroni correction, i.e., alpha = 0.05/# of CpGs) correlations in all three comparisons. These CpGs were included in two subsequent investigations. First, a query of the EWAS Atlas database was performed using these overlapping CpGs to identify meaningful associations with various phenotypes, pathways, and genomic locations [20]. Second, a query of the BBMRI-omics atlas for associated mQTLs (previously reported by Bonder et al., 2017) was performed with these overlapping CpGs to investigate their previously identified genetic relationships [21].

## 5. Conclusions

Over the past decade, the amount of research performed utilizing array-based technologies to study DNA methylation has grown rapidly, in part because of the advances in technology that enable assessments on an epidemiological scale. This also enables the next phase of studies, namely, longitudinal and meta-analysis projects. We contribute one of the first studies with new information on the stability of these measurable CpGs both longitudinally and across whole-blood and buccal samples. Our analyses identified over 7000 CpGs that showed strong correlations across blood and buccal samples. These correlated CpGs were more often associated with mQTLs, indicating that the effects of genetic variation are one of the factors that influence those areas of the methylome where inter-individual differences are conserved across samples of differing tissue sources. Further, we also provided a list of CpGs with strong longitudinal correlations in blood samples from adults over more than 10 years. We believe these CpGs to be informative indicators of longitudinally stable inter-individual differences. Finally, we provided a list of CpGs that were consistently measured across the Illumina 450K and EPIC platforms, which serves as a valuable asset for researchers considering performing analyses combining data from multiple Illumina array platforms. Subsetting to a selective list of CpGs with strong correlations across 450K and EPIC platforms (like the ones identified here or in Sugden et al.) is likely to improve the replicability of results and reduce multiple testing [18]. This information can be incorporated into future studies in order to achieve increased power.

## Figures and Tables

**Figure 1 ijms-24-14640-f001:**
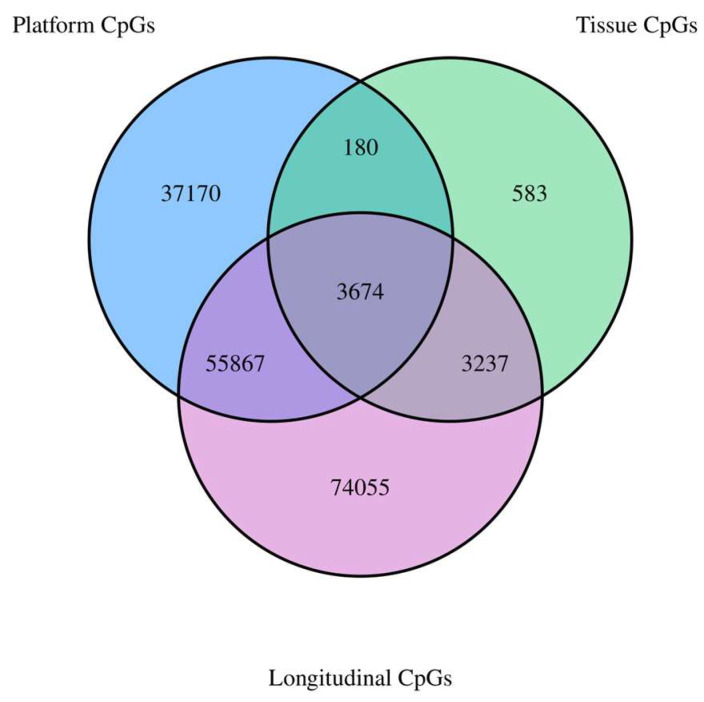
Venn diagram showing the overlap of strongly correlated (r > 0.50) and significant CpGs across the three experiments.

**Figure 2 ijms-24-14640-f002:**
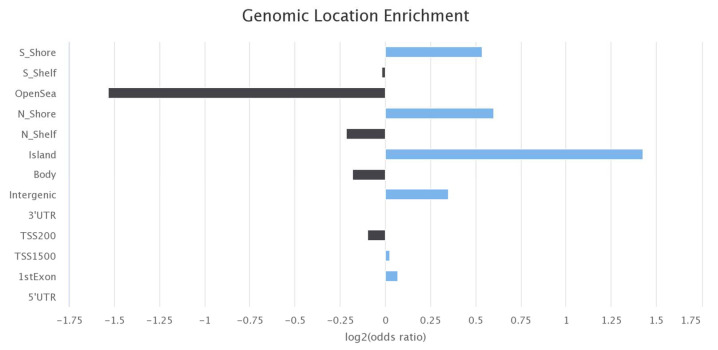
Results and figures from the query of the EWAS Atlas [20]. This is a bar plot showing the genomic locations that are associated with the identified overlapping 3674 CpGs (blue bars indicate positive enrichment and black bars indicate negative enrichment).

**Figure 3 ijms-24-14640-f003:**
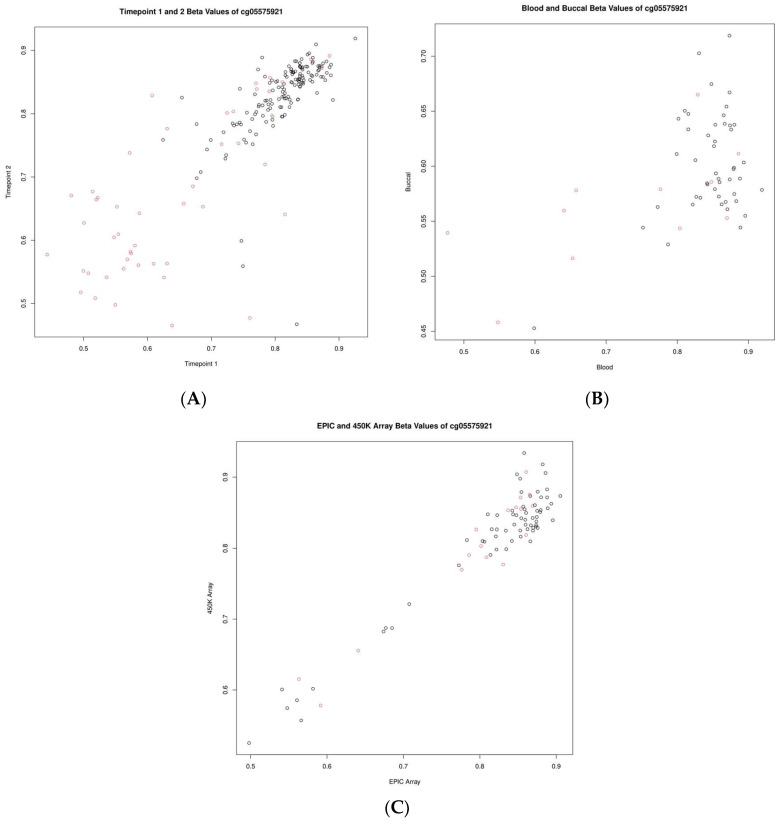
Scatter plots showing the relationship of beta values for the CpG cg05575921 (*AHRR*). (**A**) Between timepoints, (**B**) between blood and buccal samples, and (**C**) between measurements of differing platforms (EPIC vs. 450K). Smoking status is indicated by the color of the circle; grey indicates non-smokers and red indicates smokers.

**Figure 4 ijms-24-14640-f004:**
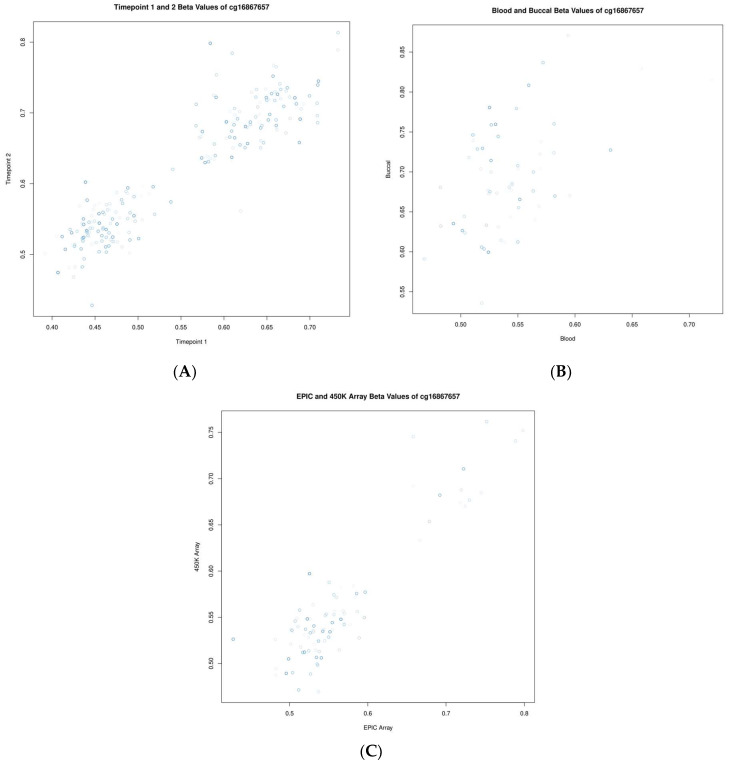
Scatter plots showing the relationship of beta values for the CpG cg16867657 (*ELOVL2*). (**A**) Between timepoints, (**B**) between blood and buccal samples, and (**C**) between measurements of differing platforms (EPIC vs. 450K). Each point represents one individual, and the shading of that point from light to dark indicates the individual’s relative age within the group (from young to old, respectively).

**Table 1 ijms-24-14640-t001:** Study populations and CPGs included for each experiment.

Experiment	Individuals	Males	Females	CpGs
Longitudinal	197	95	102	759,263
Tissue	58	21	37	759,263
Platform	83	36	47	386,805

**Table 2 ijms-24-14640-t002:** Strong (>0.50) and significant CpGs of each experiment.

Experiment	Significant and Strongly Correlated CpGs
Longitudinal	136,833
Tissue	7674
Platform	96,891
Overlapping	3674

**Table 3 ijms-24-14640-t003:** Shown here are the 10 traits that are most significantly and abundantly associated with the 3674 overlapping CpGs identified through the query of the EWAS Atlas database [20]. The column “Background” indicates the total number of CpGs that were identified in previous studies to associate with that particular trait and are represented in the EWAS Atlas [20].

Trait	Odds Ratio	*p*-Value	CpGs Identified	Background
Ancestry	8.887	0	328	10,618
Kabuki syndrome (KS)	25.596	3.18 × 10^−306^	162	1891
Respiratory allergies (RAs)	78.282	3.43 × 10^−296^	109	485
Alzheimer’s disease (AD)	19.434	4.39 × 10^−160^	94	1392
Gestational diabetes mellitus	6.541	1.50 × 10^−135^	156	6599
Ankylosing spondylitis	60.141	1.25 × 10^−105^	41	222
Childhood stress	26.544	2.17 × 10^−98^	50	550
Primary Sjögren’s syndrome (pSS)	7.565	3.88 × 10^−96^	97	3526
Klinefelter syndrome	64.282	3.79 × 10^−92^	35	179
Leukoaraiosis (LA)	25.757	1.50 × 10^−91^	47	531

**Table 4 ijms-24-14640-t004:** Summary of mQTL investigation of identified CpGs in each experiment.

	CpGs	mQTL Associations	Cis/Trans-mQTL Breakdown	CpGs with At Least 1 Association	CpGs Cis/Trans-mQTL Breakdown
Bonder et al. [21]	405,709	299,853	272,037/27,816	145,792 (35.9%)	142,126 (35.0%)/10,141 (2.5%)
Longitudinal	69,570	121,253	107,601/13,652	42,881 (61.6%)	41,609 (59.8%)/4662 (6.7%)
Tissue	4408	7402	6613/789	2181 (49.5%)	2119 (48.1%)/168 (3.8%)
Platform	96,891	165,484	146,330/19,154	61,503 (63.5%)	59,561 (61.5%)/6849 (7.1%)
Overlapping	3674	7016	6235/781	1989 (54.1%)	1932 (52.6%)/163 (4.4%)

## Data Availability

The HumanMethylation450 BeadChip data from the NTR are available as part of the Biobank-based Integrative Omics Studies (BIOS) Consortium in the European Genome-phenome Archive (EGA) (accession code: EGAD00010000887). They are also available upon request via the BBMRI-NL BIOS consortium (https://www.bbmri.nl/acquisition-use-analyze/bios (accessed 20 August 2023)). All NTR data can be requested from bona fide researchers (https://ntr-data-request.psy.vu.nl/). Because of the consent given by the study participants, the data cannot be made publicly available. The pipeline for DNA methylation–array analysis developed by the Biobank-based Integrative Omics Study (BIOS) consortium is available at https://molepi.github.io/DNAmArray_workflow/ (https://doi.org/10.5281/zenodo.3355292).

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
