# Peer review of "Genome-Wide DNA Methylation Profiles in Whole-Blood and Buccal Samples—Cross-Sectional, Longitudinal, and across Platforms"

_ijms, 2023, doi:10.3390/ijms241914640_

Round 1

Reviewer 1 Report

This paper presents an extremely large amount of data and is rich in useful information. Similar approaches in the past are correctly introduced, and I think this is a nice piece of work.

 I have one comment on this manuscript.; the conclusion part should be improved.

Authors should organize and explain the new facts revealed by the analysis of this paper in terms with biological significance.

There is a spelling mistake in line 54.  The word ascocciation? This should be changed to association

Author Response

Regards,

Austin Van Asselt

Reviewer 2 Report

In this manuscript, Van Asselt and colleagues described a comprehensive comparative analysis of genome-wide DNA methylation data across time (longitudinal experiment, two time points ~10 years apart), platform (the 450K vs. the EPIC Illumina arrays), and cross-tissue (for two highly explored tissues in the context of DNA methylation: whole blood vs. buccal samples).

I found the manuscript interesting and well-written. Please find below my comments, which I hope will be helpful in improving it.

1)     I couldn’t find the number of male and female participants. Just mention that the numbers “are roughly equal,” please specify.

2)     Did the authors use any deconvolution to account for the cell-type heterogeneity of the samples?

3)     The authors refer several times along the text to the BBMRI mQTL database and to Bonder et al.

Similarly, they refer to the EWAS Atlas. For both resources, there are no references to the discussed databases. I could find them only in the Methods section, which is at the bottom of the text. I suggest the authors refer to the relevant papers upfront the first time they are mentioned in the text. Also, adding a link to the database may help the readers.

4)     Same for “Sugen et al.” with no corresponding reference number.

5)     There are 10 supplementary figures. They provide valuable information. However, I suggest rearranging them. The authors refer to supplementary figure 10; later on in the text comes supplementary Figures 3,1,4,2 ... in this order, which is very confusing.

6)     In addition, supplementary figures 5-9 are not mentioned in the text at all. Also, the figure legend of some of the supplementary figures has (A), (B)… labels that don’t appear next to the panels in the figure.

7)     I think that it is not stressed enough (only one short sentence at the end of the text) that the data strongly supports focusing on subsets of highly correlated CpGs in future studies that will aim to compare cross-tissue and \or cross-platform data.
Personally, I found it the most interesting part and valuable part of the manuscript !

Author Response

Best Regards,

Austin Van Asselt
